# Metasurfaces for Sensing Applications: Gas, Bio and Chemical

**DOI:** 10.3390/s22186896

**Published:** 2022-09-13

**Authors:** Shawana Tabassum, SK Nayemuzzaman, Manish Kala, Akhilesh Kumar Mishra, Satyendra Kumar Mishra

**Affiliations:** 1Electrical Engineering, The University of Texas at Tyler, Tyler, TX 75799, USA; 2Department of Physics, Indian Institute of Technology Roorkee, Roorkee 247667, India; 3Centre of Optics and Photonics (COPL), University of Laval, Quebec, QC G1V 0A6, Canada

**Keywords:** metasurface, plasmonics, gas sensor, biosensor

## Abstract

Performance of photonic devices critically depends upon their efficiency on controlling the flow of light therein. In the recent past, the implementation of plasmonics, two-dimensional (2D) materials and metamaterials for enhanced light-matter interaction (through concepts such as sub-wavelength light confinement and dynamic wavefront shape manipulation) led to diverse applications belonging to spectroscopy, imaging and optical sensing etc. While 2D materials such as graphene, MoS_2_ etc., are still being explored in optical sensing in last few years, the application of plasmonics and metamaterials is limited owing to the involvement of noble metals having a constant electron density. The capability of competently controlling the electron density of noble metals is very limited. Further, due to absorption characteristics of metals, the plasmonic and metamaterial devices suffer from large optical loss. Hence, the photonic devices (sensors, in particular) require that an efficient dynamic control of light at nanoscale through field (electric or optical) variation using substitute low-loss materials. One such option may be plasmonic metasurfaces. Metasurfaces are arrays of optical antenna-like anisotropic structures (sub-wavelength size), which are designated to control the amplitude and phase of reflected, scattered and transmitted components of incident light radiation. The present review put forth recent development on metamaterial and metastructure-based various sensors.

## 1. Introduction

A single event has never defined the emergence of a new and emerging field of science. This is also true for metamaterials, a field that has gradually accumulated knowledge through consistent and dedicated research over the past century. A major factor in the development of antenna was technologies related to wireless communication. The scalability and efficiency of these antennas and the simplification of underlying physical modelling have great advantages over isolated antennas, such as reducing their size to that of the wavelength of the light. Natural optical devices control the wave front of light such as polarization, phase and amplitude. According to classical optics, atoms and molecules composing the medium shape the behaviour of light in naturally occurring materials. As a result of refractive index differences in the media, refraction, reflection and diffraction can all be controlled. However, natural materials tend to have small deviations in their properties when manipulated optically [1,2,3,4,5]. Various types and configurations of chemical, bio, gas and refractive index optical sensors have already been reported. There are advantages to both fibre-based and waveguide-based sensors. Some SPR and LSPR sensors are growing rapidly and opening up a lot of possibilities [6,7,8,9,10,11,12]. Through the integration of metasurface, a whole new world of senses can be opened up. Sensitivity can be enhanced, detection accuracy can be improved and the size can be compacted.

Metamaterials are subwavelength periodic metallic and dielectric structures, exhibiting properties that cannot be found in nature, which couple to the electric and magnetic components of incident electromagnetic fields. Over the past 15 years, this micro- and nano-structured artificial media class has attracted considerable attention and produced ground-breaking electromagnetic and photonic phenomena. Despite their potential, however, the high losses and strong dispersion associated with resonant responses and the use of metallic structures and the difficulties of fabricating 3D structures at the micro-and nanoscale have largely prevented the effective use of metamaterials. Through lithography and nanoimprinting, it is possible to manufacture planar metamaterials and metasurfaces with subwavelength thickness. Wave reflection losses can be greatly reduced by applying a very thin layer in the wave propagation direction. With metasurfaces, optical wavefronts can be modulated into any desired shape, and functional materials can be integrated to accomplish various objectives (e.g., altering amplitude, phase, polarisation). Moreover, nonlinearity is greatly enhanced and enables active control. There has been increasing interest in 2D planar metamaterials, namely metasurfaces. They can provide many of the same phenomena as metamaterials, except that they are a fraction of a wavelength thin, easier to fabricate, theoretically simpler to realise, and have negligible losses. It has been used to realise many optical devices. Usually, metasurfaces engineer the wavefront of light by abrupt phase changes [13,14,15,16,17,18].

Aside from superlensing, slow light and cloaking devices, refractive index (RI) bio-sensing is the most realistic and representative application of them all. A change in the RI results from biomolecular interactions occurring in analyte layers. Sensors such as the electromagnetic (EM) RI can be used in a variety of chemical and biological sensing applications due to their unique capabilities for sensitive and label-free biochemical assays. The resonant EM spectrum that is dominated by the environment can be vastly tuned by engineering individual MAs (meta-atoms) and their arrangements. This resonant property allows variation in the scattering output spectrum, which is used to measure the RI of the surrounding biomolecular analytes. Therefore, certain wavelengths and certain sensitivity levels have to be designed in mass setups. Additionally, RI sensors based on metamaterial (MM)- and metastructure (MS)-based sensing platforms have several advantages over conventional surface plasmon polariton (SPP)-based biosensors. MM- and MS-based RI sensors have superior performance than SPP-based sensors, primarily due to fabrication tolerance and signal stability, as RI variation is detected through macroscopic optical responses, mainly reflection or transmission of focused input beams [13,14]. The second advantage of periodic MAs is lower radiative damping and a higher quality factor, provided by interesting physical mechanisms such as plasmonically induced transparency or Fano resonances. A single nanophotonic RI sensor can expand its capabilities if it is combined with MM or MS. Combining multiple MAs in a unit cell or supercell can result in multiple resonances and a broad range of slow light effects, which are difficult to achieve in SPP sensors [19,20,21,22,23,24,25,26,27].

An overview of recent advances in EM MM and MS applications for various sensors is presented in this review. Sensor applications can be divided into two major categories: RI sensing with an optical response and MS sensing with the properties of light itself.

## 2. Fundamentals of Metasurfaces

MMs and MSs have centred attention of research fraternity due to their anomalous and tuneable properties. MMs are made up of periodic subwavelength metal/dielectric structures. These structures resonantly couple to electric and magnetic fields of the incident electromagnetic waves. Optical properties of MMs and MSs are decided by geometrical parameters of their constituents, called MAs. MA can be composed of one or more subwavelength sized nanostructures of noble metals or high index dielectrics. Smith [28] and Pendry [29] designed first artificial materials predicted theoretically far earlier in 1968 by Veselago [30]. After that many new exciting functionalities have been achieved in MMs such as negative refractive index, nearly perfect absorption, transmission and reflection which have potential applications in superlensing, electromagnetic cloaking etc. At present, MSs (subwavelength thick metamaterials) are replacing MMs that make it possible to achieve new applications such as planar lenses, generalisation of Snell’s law, ultrathin invisibility cloaks to name a few [31,32,33]. They are easy to fabricate and cost effective in comparison to MMs. They can give spatially varying optical responses (e.g., amplitude, phase, polarisation), which are used to manipulate wavefronts into desirable shapes. Due to their strong wavefront modulation capability in the sub-wavelength domain, various meta-devices have been introduced in recent years, such as meta-lens, absorber, vortex beam generator, holograms and many more. Generally, MSs are characterised into two classes: plasmonic (metallic) and dielectric MS. In plasmonic MSs, collective oscillations of electrons in a metal give rise to resonance, called localised surface plasmon resonance (LSPR). Plasmonic MSs have advantages such as the ability to sense analytes directly at the metal surface where field is confined strongly. This intense field confinement enhances the light matter interaction with the analyte which strongly alters the spectral response. These exciting properties make MSs a prominent candidate for sensing applications. However, metals offer significant joule heating which can alter the property of the analyte. Furthermore, high dissipation can also give rise to low quality factor (Q-factor) in a resonator. Q-factor is a measure of the energy stored in the resonator relative to the energy lost in radiation or joule heating. Low Q-factor limits the detection sensitivity. To resolve the loss issue, MSs are designed using dielectric nanoparticles which support electric and magnetic modes based on the Mie theory. Dielectric MSs have larger Q-factor in comparison to plasmonic MSs due to the absence of joule heating. However, the modes supported by dielectric MAs are less localised and have larger mode volume. For the sensing applications, dielectric MSs could be advantageous if large analyte volume is being used. Whenever MS is illuminated with a broad light source, the wavelength corresponding to the resonant wavelength is reflected due to the strong scattering, while the other wavelengths will pass through. When the incident light coincides with the resonant wavelength, the near fields of the MSs are increased in accordance with the Q-factor of the resonance. Therefore, the interaction between incident light and analyte will enhance. Q-factor can be improved via Fano resonance. Fano resonance is a type of resonance which results in asymmetric line-shape. This asymmetric line-shape is due to interference between two scattering amplitudes, one lies in the continuum state and the other lies in the discrete state. In 2007, N. I. Zheludev’s group observed Fano resonance for the first time in the microwave frequency range using asymmetric split rings (acting as resonator) for MS [34,35,36,37,38,39,40,41,42,43,44]. Here, Fano resonance is achieved by breaking the symmetry of nanostructures. In subwavelength nanostructures, dipole moments are excited which usually have broad spectral response. By breaking symmetry, narrow ‘dark’ modes, which exist due to the higher order oscillations, are excited and they interact with broad ‘bright’ mode [32]. Fano resonance depends on the degree of asymmetry of the MSs and refractive index of surrounding materials. In Fano resonance, sharp resonance peaks with high Q-factor are observed. Due to high Q-factor, Fano resonance MSs are seeing immense research attention. The above discussed mechanism of MSs can be used for various optical sensing applications such as refractive index sensing, chemical sensing, bio sensing and gas sensing.

## 3. Application of Metasurfaces in Analyte Sensing

### 3.1. Bio Sensing

MMs and MSs have opened new frontiers in many research areas. In particular, in the sensing field, sensors based on these artificially engineered materials have an edge due to high sensitivity and selective detection and measurement of biomarkers exploited majorly for accurate and early diagnosis of disease conditions. MSs and MMs introduce novel functionalities to conventional plasmonic sensors by enhancing sensitivity, limit of detection and allowing low-cost fabrication, giving rise to hybrid sensing paradigm. There are two primary types of plasmonic excitations, surface plasmon polariton (SPP) and localised surface plasmon resonance (LSPR). The surface plasmon resonance (SPR) sensors have been extensively investigated over the past few decades, resulting in many research articles and several commercial implementations [45,46]. LSPR is produced by the oscillation of free electrons at confined metal (Au, Ag, Cu, Pt, etc.)–dielectric interface, such as in metal nanoparticles, upon excitation by p-polarised light [47]. Some salient features of plasmonic sensors include real-time monitoring of binding dynamics of biomarkers on the device surface, reusability, fast response, straightforward sample treatments and label-free detection at the point of care. However, conventional SPR instruments have several limitations, including a lack of multiplexing capability and hence low throughput, dependence on the specific binding surface, chemical inertness to metal surfaces leading to reduced sensitivity, lack of wireless operation and risk of data misinterpretation [48]. Typically, despite the availability of different combinations of metal and dielectric materials, substantial modulation of optical properties is not feasible, thereby lacking manoeuvrability. In contrast, composite structures such as metamaterial and MS-based structures with negative permittivity, permeability and perfect absorption, can be utilised to tailor the optical properties near the metal-dielectric interface [49]. Plasmonic MS-based sensors follow the fundamentals of optical properties near the MS-dielectric boundary. Russian physicist Victor Veselago first introduced the theoretical approach of negative refractive index (RI) material in 1968 [28]. The MM-based RI sensor was experimentally demonstrated at microwave frequency in 2000 [31]. The MM-based plasmonic biosensor has been successfully implemented in 2D and 3D nanostructures for different bio-analyte detection. These sensors have drawn much attention due to their ultrahigh sensitivity compared to conventional plasmonic biosensors [50]. Plasmonics and their meta configurations have been utilised to detect a variety of viruses, including hepatitis B [51], Zika Virus [52], HIV DNA [53], SARS-CoV-2 [54] and malaria [55]. This section presents a review of recent meta-surfaced and meta-structured sensors for biosensing applications. This section is subdivided into subsections based on the type of metastructure/metasurfaces used for biosensing applications. The advantages and disadvantages (where applicable) of various biosensors are then compared in Table 1 at the end of this section.

#### 3.1.1. LC Resonator-Based Biosensors in THz/GHz Regime

Plasmonic MSs are composed of novel metals and tailoring their optical and geometric properties will allow the detection of biomarkers and biomolecules with high sensitivity. Such structures that operate in the terahertz/gigahertz (THz/GHz) frequency region, are suitable for label-free and contactless sensing of different viruses, bacteria and cancer biomarkers. The MM-based plasmonic sensor is generally modelled as an LC circuit, where L stands for inductance and C stands for capacitance. The target analytes and biomolecules bind to the nanoscale gaps of the MS, exhibiting a shift in the resonance frequency. This shift in frequency/wavelength depends on the counts, concentration and binding properties of biomolecules at the metallic surface. For instance, a MM-based plasmonic sensor was proposed by Lee et al. to detect biotin and streptavidin. A split ring resonator (SRR) was fabricated on a printed circuit board (PCB) with copper (Cu), nickel (Ni) and gold (Au) as the metals [56]. A surface electric current was generated on the LC resonator surface upon excitation by a time-varying magnetic field. Biotin and streptavidin molecules bound covalently to the metallic gap, manifesting as a change in the capacitance of the LC resonator and consequently resulting in a resonance shift proportional to the concentration of the biomolecules. The observed resonance shifts were 120 MHz and 40 MHz for biotin and streptavidin, respectively. 

Kashiwagi et al. proposed a similar THz metamaterial made with an array of SRRs, using inkjet printing of silver nanoparticles [57]. Inkjet printing provided a simple, robust and cost-effective way of fabricating the SRR arrays on flexible paper and plastic substrates, as is shown in Figure 1a–c. In this work, six different designs with line widths of 50 µm, 100 µm and 150 µm, and capacitive gaps of 50 µm and 100 µm were fabricated and analysed. A total of 48 SRR sensors were fabricated per six different SRR patterns. The fabrication error was minimal for 100 µm and 150 µm line widths and 100 µm capacitive gap. The resonance frequencies were observed at 0.27 THz and 0.37 THz using THz time-domain spectroscopy (THz-TDS) characterisation. Figure 1d,e demonstrates the transmission spectra of the SRRs for x- and y-polarised incident waves. A good agreement was observed between the simulation and experimental results, as is evident from the transmission spectra shown in Figure 1d,e. This multi-resonance sensor can be utilised for low-cost biosensing applications such as viruses, proteins and other biomarker detection. 

Further, Tao et al. proposed a cost-effective, disposable and easy-to-use SRR sensor on a paper substrate for quantitative analysis of blood glucose, useful for on-site detection and analysis in low resource settings [58]. Three LC resonator structures were fabricated, two purely electric resonators (i.e., polarisation-sensitive and nonsensitive polarisation resonators, both with a unit cell size of 100 μm × 100 μm) and a canonical SRR (with a unit cell size of 50 μm × 50 μm). A photoresist-free shadow mask deposition technique was used to fabricate the structures. The prototype structure was tested with glucose concentrations ranging from 3 mmol L^−1^ to 30 mmol L^−1^. SRR-based sensors also play a vital role in virus detection. Park et al. proposed an SRR sensor for detecting low concentration bacteriophage viruses with sizes ranging from 60 nm (PRD1 double-stranded DNA Virus) to 30 nm (MS2 single-stranded RNA virus) [59]. Since the size of these viruses was less than the scattering cross-section of the incident THz wave, the shift in LC resonance frequency was owing to the variations in the dielectric constant of the split gap in the SRR array. Upon binding of the virus particles to the split gap, the dielectric constant was changed, which resulted in a shift in the resonance frequency, a phenomenon that was observed by the same research group in another work [60]. The SRR structure reported in [59] was fabricated using e-beam lithography followed by the e-beam deposition of thin films of gold (97 nm) and Cr (3 nm) with a linewidth of 4 µm, as shown in Figure 2. Solutions containing PRD1 and MS2 viruses with a density of 10^9^/mL were used to quantify the sensor’s sensitivity. About 10 μL of the virus solution was drop cast on a 10 mm^2^ surface area. It was observed that the sensitivity to the PRD1 virus increased from 6 GHz μm^2^/particle to 80 GHz μm^2^/particle (almost 13-fold improvement) as the gap width decreased from 3 μm to 200 nm.

The detection of microorganisms has been studied for decades. Several organisms, such as bacteria and fungi, are responsible for life-threatening diseases in humans, including tuberculosis, asthma [61], gonorrhoea and meningitis [62]. Therefore, early identification of these microorganisms is crucial to preventing the spread of infections and treating them efficiently. Park et al. reported a THz biosensor to facilitate accurate and faster detection of such organisms. A detailed overview of the device is provided in [60] and illustrated in Figure 3. The THz wave propagation through the MM sensor was analysed in response to fungi, which was bound to the metamaterial resonator split gap (length 2–3 µm). SEM images of the penicillia-coated metastructure are portrayed in Figure 3a,b. Furthermore, the work was extended by detecting the *E. coli* bacteria in an aqueous medium. Figure 3c denotes a schematic of the experimental setup and Figure 3d shows the percent transmission change in different media. Moreover, antibody functionalisation was found to be crucial for improved sensitivity. Figure 3e,f depicts the transmission spectra with and without anti-*E. coli* functionalisation on the sensor surface. The shift in resonance frequency was minimal without anti-*E. coli* functionalisation [60].

Additionally, Chen et al. demonstrate an active metamaterial device capable of efficient real-time control and manipulation of THz radiation [63]. Li et al. proposed high-sensitive LC resonator structures to measure the high molecular weight protein BSA (bovine serum albumin) [64]. The reported biosensor had four identical LC resonators in a unit cell and was fabricated using a surface micromachining process with aluminium as the metal layer. The fabricated sensor had an overall dimension of 10 mm × 10 mm. The sensitivity of the biosensor was found to be 85 GHz/RIU, and a maximum resonance frequency shift of 50 GHz was observed at a concentration of 765 μmol/L. The sensor was capable of measuring the BSA concentration down to 1.5 μmol/L. The high sensitivity was achieved owing to the four identical LC resonators, which outperformed a single resonance-based LC structure [64].

The sensitivity of MM-based THz biosensors can be further enhanced by adding gold nanoparticles (AuNPs) to a metal film. In this regard, Liu and his colleagues provided an efficient approach to increasing sensitivity to EGFR (epidermal growth factor receptor) in the THz regime using a bow-tie metamaterial array [65]. EGFR is critical in the diagnosis and prognosis of tumour tissues, cancer cell-like gastrointestinal cancer, lung cancer and oral squamous cell carcinoma regrowth rate [66,67]. The proposed MM-based biosensor reported in [65] had a bow-tie structure comprising an array of chromium (20 nm) and gold (Au) (100 nm) bilayer films coated with AuNPs, which provided an extraordinary sensing performance. The geometrical design along with the chemical functionalisation on the metamaterial structure are depicted in Figure 4a,b. Furthermore, shifts in the resonance frequency were analysed for different AuNP diameters. It was evident that increasing the diameter of the AuNPs enhanced the resonance frequency shifts. Figure 4c–e demonstrate the transmission spectra of the bow-tie structure in response to varying concentrations of EGFR when coated with AuNP sizes of 5 nm, 15 nm and 25 nm. The frequency shift increased with increasing AuNPs size when the sensor was functionalized with antibody. Figure 4f portrays the frequency shifts with EGFR concentrations with and without antibody functionalisation for different AuNPs size. The authors suggested that the sensitivity could be tailored by optimising the MM design with a low absorption substrate such as a thin film of silicon nitride instead of the bulk Si substrate [68]. Besides protein detection, this design can be also reconfigured for detecting cancer cells and bacteria [65]. A different type of bow-tie plasmonic antenna apertures was also reported in [69].

The water absorption bands in the THz regime (the translational mode centred around ~6 THz and rotational modes spanning from 10–20 THz) interfere with the detection of low concentration analytes such as protein, virus and bacteria in a microfluidic channel, thereby necessitating the reconfiguration of the sensing structure or selection of appropriate wavelength regime [70]. Geng et al. integrated a microfluidic chip with an SRR-based metamaterial sensor to detect Alpha-fetoprotein (AFP) and glutamine transferase isozymes II (GGT-II) for early diagnosis of liver cancer [71]. Figure 5a,b depict the schematic of the proposed MM sensor integrated with a microfluidic chip and its equivalent electrical circuit modelling with inductance and capacitance. Fabrication steps of the SRR structure included RCA standard cleaning, lithography, deposition and lift-off, followed by the deposition of a 200 nm-thick gold metallic layer using radio magnetron sputtering. Afterward, a polydimethylsiloxane (PDMS) microfluidic channel was introduced to the surface in order to control the sample volume needed for surface functionalisation. The results at each fabrication step are highlighted in Figure 5c–g. The PDMS channel was peeled off prior to the THz testing owing to the low transmission of PDMS in THz (less than 50%) and high absorption of THz energy by water. The proposed design was analysed for three different gaps in the SRRs (gap widths of 2 µm, 4 µm and 6 µm). The characteristics of the sensing performance were determined by FDTD simulation. With an increase in the refractive index of the ambient medium, there was a blueshift in the resonance frequency. Furthermore, an asymmetry was introduced to increase the Fano resonance of the SRR sensor, which had a significant impact on the Q factor. Total resonance shifts of 19 GHz and 14.2 GHz were observed for 5 μm/mL of GGT-II antigen and 0.02524 μg/mL of AFP, respectively, with the SRR having two gaps. The sensitivity was found promising for detecting cancer biomarkers.

#### 3.1.2. Hyperbolic Metamaterials (HMMs)

SPR and LSPR sensors require bulky optical elements, including a prism, unsuitable for label-free, point-of-care applications. LSPR, SPR and SRR-based metamaterial sensors generally exhibit sensitivity around $\approx 2\times 10^{2}\frac{\mathrm{nm}}{\mathrm{RIU}}$ [72,73]. Although grating-coupled SPR sensors are not bulky, the sensitivity is lower than prism-coupled SPRs. Hence, it is very challenging to detect highly diluted biomarkers. Recently, nanorod MMs showed ultra-high sensitivity with promising biosensing applications [74]. Sreekanth et al. demonstrated a plasmonic biosensor based on HMMs with sensitivity in visible and near-infrared regions of the electromagnetic spectrum [75]. The HMMs were coupled with 2D gold diffraction gratings, with an advantage of high tunability in the sensitivity and mode of operation (from visible to NIR) achieved by modulating the geometry and grating parameters. The structure was composed of 16 alternating layers of gold and Al_2_O_3_ (10 nm) thin films [68]. A metallic layer was added to the HMMs to diffract the incident wave that produced a wide range of wavevectors inside the HMMs. The structure was demonstrated to detect different concentrations of BSA and biotin ranging from 10 pM to 1 µM. One significant finding from this work is the nonlinear variation of the wavelength shifts with analyte concentrations. The sensor could detect low molecular-weight (244 Da) molecules down to picomolar levels. The highest sensitivity of this proposed biosensor was 30,000 $\frac{\mathrm{nm}}{\mathrm{RIU}}$ with a figure of merit (FOM) of 590. FOM is defined as (Δλ/Δη) (1/Δω), where Δλ is the wavelength shift, Δη is the change in the RI of the analyte medium and Δω is the full width at half-maximum at the resonance wavelength. In conclusion, hyperbolic metamaterials have considerable prospects in biosensing owing to their ultrahigh sensitivity and ultralow detection limit.

### 3.2. Gas Sensing

Gas sensors (also known as gas detectors) are electronic devices that detect and identify gases, including CO_2_, SO_2_, NO_x_ and toxic and explosive gases. Gas sensors are employed in factories and manufacturing facilities to identify gas leaks and detect smoke and carbon monoxide emissions at home. Japan implemented the first semiconductor oxide-based gas sensors in the 1970s [76], current-type oxygen sensors [77] and ceramic humidity sensors for automatic cooking ovens [78]. Currently, semiconductor, electrolyte or catalytic combustion type sensors detect gases such as methane, propane, carbon monoxide, ammonia, hydrogen sulphide, oxygen, nitrogen dioxide, ozone, etc. These gas sensors are used in safety industries for the detection of explosives [79], indoor air quality/HVAC industries, medical and life-science industries [80,81], aerospace industries, agriculture industries [82], modified atmosphere packaging (MAP) industries, transportation industries, fire suppression testing, university research applications and many more. Different types of sensors have already been studied for various applications and schemes, such as acoustic gas sensors, carbon nanotube (CNT) sensors, catalytic gas sensors, electrochemical gas sensors, thermal conductivity-based gas sensors, optical gas sensors, metal oxide semiconductor (MOS)-based gas sensors, organic chemiresistive gas sensors, piezoelectric gas sensors, photonic crystal-based gas sensors, micro-electro-mechanical systems (MEMS) and metamaterial absorber systems [83]. The electrochemical sensors can detect a wide range of gases at low concentrations. Although optical fibre-based gas sensors provide dynamic monitoring with high repeatability and reusability, they are still susceptible to ambient light interference [84]. In contrast, the semiconductor gas sensors are mechanically robust [85] but exhibit nonlinear responses under environmental variations such as humidity changes [86].

One approach to improving gas sensing performance is manufacturing artificially engineered MM absorbers [87]. MM-based sensing can be realised in the microwave, terahertz (THz), infrared (IR), visible and ultraviolet (UV) regimes. Due to controllable optical parameters, the performance is enhanced by introducing plasmonics in a metal-insulator-metal (MIM) structure [88]. Planar MIM (p-MIM) and vertical MIM (v-MIM) structures are reported in the literature, but these structures inhibit interaction of the target analyte with the hot spot region [89]. A vertically oriented channel MIM (c-MIM) structure was proposed by Su et al. to overcome this limitation [90], where a plasmonic molecular region (hot spot region) was introduced to provide enhanced sensitivity. This c-MIM structure was demonstrated to detect carbon dioxide and butane gases. The higher sensitivity resulted from the presence of a gap between the metal conductors. As a result, s-excited polaritons were coupled in the gap, and this phenomenon is called channel plasmon polaritons [91]. Furthermore, Fano-like resonance was evident due to the combined response from plasmonic resonance and the molecular vibration effect, which improved the sensitivity [90]. The device could detect butane gas down to 20 ppm. However, the current Occupational Safety and Health Administration (OSHA) permissible exposure limit for n-butane is 800 ppm as an 8-h time-weighted average [92].

Much work has been conducted on mid-infrared gas detection using optical sensing modality. For example, a metamaterial perfect absorber (MPA)-based CO_2_ gas sensor was reported with a sensitivity of 22.4 ± 0.5 ppm·Hz^−0.5^ [93]. Figure 6a illustrates the metamaterial thermal emitter, and Figure 6b shows the schematic design of the unit cell of the LC resonator. Figure 6c,d denote the detector chip and the SEM image of the sensor cell.

This section is subdivided into subsections based on the type of metastructure/metasurfaces used in gas sensing applications. Owing to the limited literature available for metastructure-based gas sensors, this section is focused on CO_2_, NO_2_ and H_2_ gases, where CO_2_ is a potent greenhouse gas, NO_2_ is a toxic pollutant that contributes to producing tropospheric ozone and H_2_ that is a highly flammable gas. Finally, the performances of various gas sensors are compared in Table 2 at the end of this section. Moreover, a critical analysis of advantages and disadvantages (where applicable) of each sensor technology is provided.

#### 3.2.1. Metamaterial Perfect Absorber/Emitter

Global warming is increasing at an alarming rate, and the primary antagonist that is responsible is CO_2_ [94,95]. Additionally, the CO_2_ flow control needs to be maintained in health science and industry operations. According to a probability analysis, the long-term limit for CO_2_ lies within 300–500 ppm for 25 percent risk tolerance [96]. Therefore, novel gas sensors must be designed to detect CO_2_ levels in this range and beyond. Toward this end, a MS-based perfect absorber design was reported to detect CO_2_ at its signature absorption band of 4.26 µm [97]. The sensor was fabricated by depositing a perfect absorber comprised of silicon nano-cylindrical meta-atoms (MAs) on a gold layer [97]. Moreover, a thin layer of polyhexamethylene biguanide (PHMB) polymer was utilised as the CO_2_ gas absorption layer on top of the MS. The underlying physics is the absorption and desorption of CO_2_ gas molecules onto the PHMB layer, resulting in a corresponding change in the RI of the polymer layer. Subsequently, there was a blue shift in the resonance wavelength. Figure 7a,b depict the sensor structure and reflection spectra for increasing concentrations of CO_2_, respectively. It was evident that there was a blue shift in the resonance wavelength with increasing gas concentrations, resulting in a sensitivity of 17.3 pm/ppm. Moreover, the sensor had a wide detection range from 0–524 ppm. Although these results were based on FDTD-based numerical simulation, the authors proposed a fabrication scheme as shown in Figure 7c.

With the advent of CMOS and MEMS technology, MM perfect emitter (MPE) structures brought several folds of improvement to gas sensing [98]. CO_2_ gas was detected in the range from 0–50,000 ppm, with a five-fold increase in relative sensitivity as compared to the conventional blackbody emitter. Moreover, the proposed sensor exhibited a resonance quality factor of 15.7 at the centre wavelength of 3.96 μm for CO_2_ sensing. Consequently, the reported MM sensor was the first CMOS-compatible MPE structure with a high-quality factor for NDIR (non-dispersive infrared) gas sensing applications. Additionally, the fabrication process followed standard approaches such as thermal evaporation of the Cu backplane, atomic layer deposition (ALD) of the Al_2_O_3_ spacer layer and lift-off and dry etching with negative tone resist. Recent advancements in NDIR gas sensing have paved the way for a better detection range with higher sensitivity. For instance, a metamaterial emitter-based NDIR sensor demonstrated a four-fold enhancement in the emission intensity as compared to a standard non-plasmonic emitter [99].

Most of the previously mentioned work for gas sensors did not thoroughly investigate the selectivity, a crucial figure of merit for gas sensors. Considering the limitation above, Hasan et al. reported a hybrid CMOS metamaterial absorber with high selectivity in the mid-IR region [100]. Metamaterial absorber type NDIR sensor converts light to heat energy and can achieve high wavelength selectivity with controlled light-matter interactions. The sensor fabrication started with an 8″ silicon wafer patterned into the metamaterial structure by deep UV photolithography. Afterward, the gas-selective layer was spun coated on the wafer at varying speeds to form different thicknesses. The geometric and chemical composition of the supercell structure is depicted in Figure 8a–c. The simulation results in Figure 8d–h show no near-field interaction among the unit cells. This hybrid metamaterial absorber offered a fast response and minimal hysteresis. Additionally, dual-mode sensing using coupling and non-coupling region of operation provides an efficient technique to use the gas-selective polymer at its fullest potential for both low and high concentrations of CO_2_ gas. The steady-state response in Figure 8i indicates the maximum signal output of the sensor in response to a particular concentration. The authors suggested that increasing the effective sensing area could increase sensitivity at low concentrations to improve the overall performance. Additionally, sensitivity can be enhanced through engineering the gas-selective layer and controlling the metamaterial and absorber design to ensure better light–matter interactions.

Multiplexed monitoring of gases and classification of gas components in a mixture is essential to cancel out the cross-sensitivity to interfering gases. Such sensors find numerous applications in environmental monitoring and industry operations. For instance, in an agriculture field, fertilisers are decomposed into potent greenhouse and toxic gases, including NH_3_, NO, N_2_O, NO_2_ and CH_4_, via ammonification and nitrification processes [101]. These gases have a substantial negative impact on the environment as they are the major contributors to global warming. On the other hand, measuring and quantifying the gases emitted in agricultural land is essential for optimising the application of agrochemicals, which will lead to resource management while also reducing environmental pollution. The mid-IR regime (e.g., from 2–20 μm) is preferred for detecting multiple gases due to the molecular absorption signatures of individual gas molecules. This broad wavelength regime is called the molecular fingerprint region [102]. A multiplexed NDIR gas sensing platform consisting of a narrowband infrared detector array integrated with plasmonic metamaterial absorbers (PMA) was reported to detect multiple gases [103]. This sensor suite exhibited a detection limit of 489, 63, 2, 11, 17, 27, 54 and 104 ppm for H_2_S, CH_4_, CO_2_, CO, NO, CH_2_O, NO_2_ and SO_2_, respectively. The PMA integrated pyroelectric elements resolved different gas absorption levels. The geometrical properties and the metamaterial plasmonic resonance were tailored to identify individual gas absorption levels from their absorption spectra. Figure 9 shows the working principle and structure of the multiplexed gas cell with multiplexed gas absorption spectra for CO, NO, NO_2_, CH_2_O, NO_2_, H_2_S and SO_2_.

#### 3.2.2. Complementary Split-Ring Resonator

NO_2_ gas is one of the most toxic gases that critically damages the environment. NO_2_ exacerbates respiratory diseases such as asthma [104] from a medical perspective. From a global warming perspective, NO_x_ is also responsible for the depletion of the ozone layer [105]. Consequently, detecting the biohazardous oxides of nitrogen is crucial yet challenging [106]. Gas sensors with high sensitivity, quality factor, reasonable detection limit, fast regeneration and high selectivity are being researched to detect NOx gases. A Fe_3_O_4_-mediated complementary split-ring resonator (CSRR)-based MM structure was reported [107]. The proposed metasurface comprised of two square ring-shaped slots with the resonant frequency centred at 430 MHz and an additional signature around 300 MHz. The CSRR metamaterial structure was laid on an FR-4 epoxy (ɛ_r_ = 4.4, thickness = 0.8 mm) substrate. The device could detect NO_2_ concentrations ranging from 0–110 ppm. The gap in the copper CSRR was functionalised with the Fe_3_O_4_ nanoparticles to trap the NO_2_ molecules leading to a variation in the dielectric property and hence capacitance. The sensor demonstrated a sensitivity of 0.2 MHz/ppm with fast regeneration and good repeatability.

#### 3.2.3. Metal-Insulator-Metal

Hydrogen is a renewable energy source for the next generation of energy-efficient industries yet is considered an explosive gas [108]. H_2_ has a high-risk explosion factor, and nearly 4 to 75% is deemed explosive [109]. Hence, close monitoring of the sub-ppm level of H_2_ is essential. Optical sensors are preferred over electrical sensors for H_2_ sensing due to the spark ignition risks. Plasmonic gold nanorod, nanoparticles-based devices have drawn much attention to H_2_ gas sensing. Pd metal has a high affinity toward hydrogen. Nasir et al. reported a bimetallic Au/Pd nanorod-based sensor for monitoring H_2_ [110]. Phase change material has also been extensively studied for H_2_ gas sensing [111]. Beni et al. [112] reported a plasmonic MIM comprised of a phase change material (Y or WO_3_) sandwiched between a Pd nanodisk at the top and an Au mirror at the bottom for H_2_ gas sensing. The Y and WO_3_ materials exhibited opposite trends in phase change under H_2_ exposure [113]. For instance, the conductive Y metal became more dielectric and showed a blueshift in the reflectance dip, while WO_3_ became more metallic and demonstrated a redshift in resonance under hydrogenation. The sensor exhibited a response time of only 10 s, which is commensurate with the industry standard. In another work, aluminium-doped zinc oxide (AZO) was used for low concentration (0.7%) hydrogen detection [114]. SiO_2_ layer was deposited on top of a Si wafer, and AZO hollow nanotubes were standing on the SiO_2_ layer. Advanced ion reactive etching and atomic deposition layer (ALD) techniques were employed to fabricate the AZO hollow nanotubes. Figure 10 depicts the AZO configuration and H_2_ sensing results.

### 3.3. Chemical Sensing

One of the first MM-based chemical sensors was proposed in 2013 by Withayachumnankul et al. for simultaneously identifying methanol and ethanol at an operating frequency of 1.9 GHz [115]. Afterward, various features were incorporated, such as a microfluidic channel for detecting isopropanol, D glucose and methanol [116] and the separation of ethanol and DI water [117]. In addition to integrating the microfluidic platform, the utilisation of metamaterial as an absorber has also been proposed [118,119,120,121]. A paper-based flexible and wearable metamaterial sensor for distinguishing oil, methanol, glycerol and water is noteworthy [122]. However, these sensors suffer from a low-quality factor. Researchers have overcome this limitation by choosing substrates with reduced loss [123,124] and using different resonator design approaches [123,125]. Recently, various MM-based chemical sensors have been reported for commercial purposes [125,126]. A G-shaped resonator was developed with an improved quality factor (Figure 11 shows a comparative analysis between simulation and experimental measurements) to differentiate between pure and used transformer oil, diesel, corn oil, cotton oil, olive oil, aniline-doped ethyl alcohol and benzene-doped carbon tetrachloride [127].

Carbon nanotube (CNT)-based SPR metastructures have also been reported where the Fano model was used to optimise the sensor performance. The device demonstrated a sensitivity of 1.38 × 10^−2^/ppm from 1–10 ppm and 3.0 × 10^−3^/ppm over 10 ppm [128]. Moreover, a metamaterial-based CSRR sensor was fabricated on Roger RO3035 substrate with a thickness of 0.75 mm, a relative permittivity of 3.5 and a loss tangent of 0.0015. To improve the sensitivity and Q factor, the chemical samples were introduced to a capillary glass tube placed in parallel to the sensor surface [129]. Figure 12a,b show the geometric features of the sensor and Figure 12c–e show the physical implementation. To distinguish branded diesel oil from unbranded oil, a MM-based sensor incorporating a microstrip transmission line was developed [130]. MM-based transmission line sensor has also been used to investigate the contamination of branded local spirit by methanol [131]. The sensor demonstrated a high sensitivity to detect methanol content with a bandwidth of 150 MHz. A highly sensitive SRR metastructure integrated with a PDMS microfluidic channel has been reported for glucose monitoring [132]. An interdigitated capacitor was utilised to intensify the E field, thereby improving the sensitivity over a wide range of glucose concentrations (i.e., 0 to 5000 mg/dL) [132].

Another interesting metamaterial-based chemical sensor was developed by incorporating multiple symmetrical double SRRs. Such a structure holds promise in multi-band sensing of chemicals [133]. The core feature of this sensor lies in a miniaturised, reusable, label-free and non-destructive metamaterial-microfluidic combination to determine the chemical property of liquids. Figure 13 illustrates the theoretical electrical and magnetic field distributions, proposed design with the equivalent circuit diagram and the simulated and measured performance. Likewise, a phase change material derived from the Ge_2_Sb_2_Te_5_ (GST) combination was used to develop a temperature tuned sensor for detecting haemoglobin and urine (Figure 14) [134].

Leitis et al. developed a novel germanium-based MS that adsorbed molecules over a broad spectrum from 1100 to 1800 cm^−1^ with a substantially high Q factor [135]. This novel structure combined angle-multiplexed refractometric sensing with the chemical specificity of infrared spectroscopy, thereby eliminating the need to use complex spectroscopic equipment or tunable light sources. Figure 15 illustrates the detailed mechanism of light incident at varying angles, the result and the corresponding shift in the resonance peak and the near-field coupling between the dielectric resonators and the molecular vibrations of the analyte.

In recent years, metamaterial-based chemical sensing has drawn much attention in the MHz, GHz and THz, regimes [136,137,138,139,140]. Table 3 outlines some recently reported MM chemical sensors as well as a critical analysis of advantages of each sensor technology.

## 4. Future Trends in Metasurfaces for Sensing Applications

The rapidly evolving fields of nanophotonics have opened new frontiers in metamaterials, plasmonics and photonic crystals. Some promising recent research progress in metasurfaces include holographic displays, polarisation conversion, phase modulation and high-resolution imaging [141,142]. Recently, the combination of deep learning with nanophotonics and metasurfaces is being studied extensively. For instance, a global optimisation algorithm was developed by Jiang and Fan to train a generative neural network for inverse design of metasurfaces [143]. Such a technique can play a profound role in optimising the architecture of metasurfaces and metastructures for a variety of applications. Despite the advances in metamaterial-based structures, there is a need for low-cost and scalable production of these metastructures in order to facilitate large scale production of biomedical sensors and systems. In this regard, some promising avenues include three-dimensional (3D) printing, chalcogenide materials and hardware and software co-design. Integration of novel biomaterials with 3D printing results in cost-effective and scalable prototypes and architectures. Over the last few decades, several 3D printing techniques have demonstrated unprecedented performance in terms of producing features with micro/nano resolution. Some well-known high-resolution 3D printing methods include stereolithography [134,135,136,137,138,139,140,141,142,143,144,145,146,147], digital light processing [148], multiphoton polymerisation [149], fused deposition modelling [150,151], coaxial extrusion [152], material jetting and binder jetting [144]. With the appropriate biomaterials and 3D printing technique, it would be possible to produce next generation of metastructures with unique properties, tunable electro-optic and thermo-optic effects and analyte sensing functionalities, while also facilitating low-cost and roll-to-roll fabrication. On the other hand, from materials perspective, chalcogenide phase-change materials (that include sulphides, selenides and tellurides) have opened a new era of adaptable metastructures [153]. Particularly, germanium (Ge)–antimony (Sb)–telluride (Te) alloys exhibit large refractive index contrast (e.g., Δ*n* ≈ −1.5 at 405 nm wavelength) and almost three orders of resistivity change, which are desired features for rewritable optical disks [154,155] and electronic memories [156,157]. These materials, when embedded into the micro/nano photonic platforms, will lead to reconfigurable metastructures employed in biosensors with desired properties such as ultra-high sensitivity, high selectivity, tunable and wide dynamic range, multiplexing capability and high robustness. Seamless integration of nanophotonics platform with artificial intelligence and machine learning would enable the optimisation of design constraints such as cost, performance and power consumption of biosensors [158]. Hence, from the above discussion, it can be concluded that there are huge prospects for integrating metasurfaces with novel fabrication techniques, materials or data analytics to realise a new paradigm of bio/gas/chemical sensing.

## 5. Conclusions

In summary, we have discussed the primary applications of MS sensors in healthcare diagnosis, biomolecule detection, virus, bacteria, fungi detection, cancer detection, etc. The world is currently facing one of the life-threatening pandemics caused by the SARS-CoV-2 virus. Therefore, an early and accurate diagnosis of the virus with the help of a MM-based biosensor can prove to be a life saviour in this regard. The fascinating part of the MM sensor is its dual mode of operation, where both plasmonic, optical interaction and chemical reaction can play their role, and the sensitivity is tremendous. Additionally, the design mechanism has a feasible fabrication scheme with a low cost of operation. The chemically active layer in the biosensor can easily bind a broad range of antibodies. With the MM part, the sensitivity manifolds as compared to those of the conventional SPP-, SPR- and optical fibre-based biosensors. Recently, in situ monitoring and lab-on-chip operation have become the prime concern for researchers in the bio sensing field. With the help of embedded systems and machine learning, these sensors can collect numerous data remotely, predict future physiological conditions and prevent an outbreak of a future pandemic. Like the biosensing application, gas and chemical sensing are necessary for the growing industry in the 21st century. The produced gas is generally detected with bulky and low sensitivity optical fibre or metal-oxide-semiconductor sensors for different chemical processes in the industry. Plasmonic, CMOS absorber or emitter and MEMS technology have made MM gas sensors cost-effective and compact, with a sensitivity of around 5 to 10 times that of the conventional sensors. The industry and the environment need monitoring to detect any hazardous gases, such as greenhouse gas emissions. The MS-based gas sensor has proven effective in detecting multiple gases with high repeatability and low regeneration time (10 s Industry standard). As discussed in the application section, these MM gas sensors detect gas in a low concentration (0.7% to 2%) range. Moreover, these sensors can be easily tailored for the desired mode of operation as the geometry of the MM sensor controls the sensing characteristics. Recently, multiplexed detection [119,159,160,161] is a fascinating invention that can be adopted to MM sensors. This approach helps detect target gases in a gas mixture in the Mid-IR gas fingerprint regime. Further, this method can be improved by controlling the chemically active polymer layer, plasmonic nanoparticles and MM geometry design. Finally, we can manipulate the characteristics of MM materials to fabricate sensors as per our needs.

## Figures and Tables

**Figure 1 sensors-22-06896-f001:**
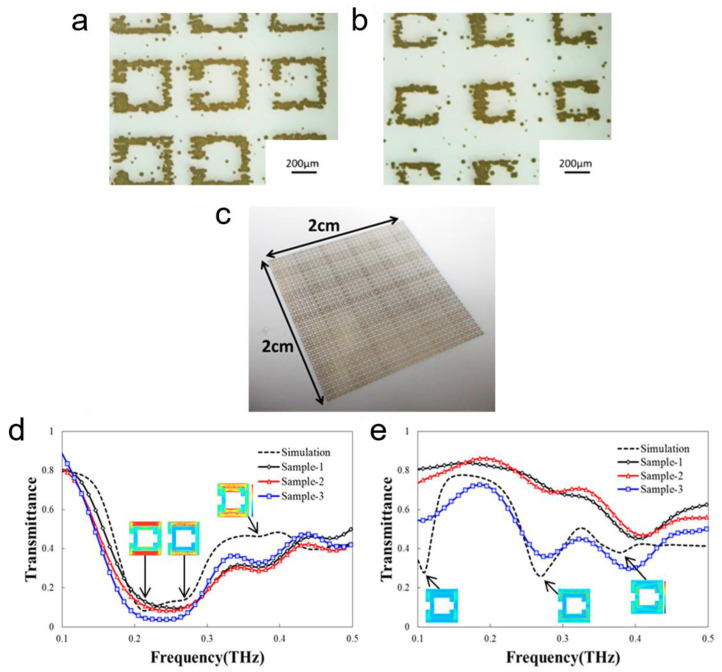
A THz Split ring resonator (SRR) fabricated by inkjet printing of silver nanoparticles. (**a**) Upper SRR array, (**b**) lower SRR array and (**c**) an overall view of the fabricated sensor. Transmission spectra of the stacked SRR arrays for (**d**) x- and (**e**) y-polarised incident waves. Insets in (**d**,**e**) represent simulated TE-modes at resonance frequencies [57]. © 2018, AIP Advance.

**Figure 2 sensors-22-06896-f002:**
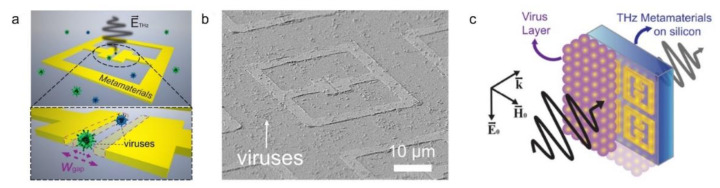
(**a**) Schematic illustration of nano-gap metamaterial structure for THz sensing of viruses. (**b**) SEM image of the viruses deposited on the structure with a gap width (w) of 200 nm. (**c**) Schematic depiction of the measurement of the dielectric constant of virus layers using the THz metamaterial structure [59] © 2017 Optical Society of America.

**Figure 3 sensors-22-06896-f003:**
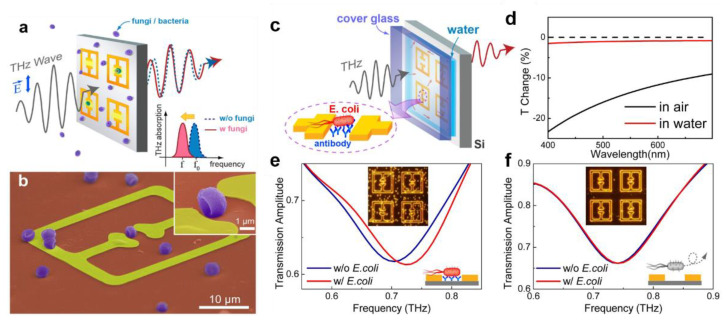
Sensing microorganisms using THz metamaterials. (**a**) A schematic representation of sensing of microorganisms with THz metamaterials. (**b**) A colour-enhanced SEM image of metamaterials coated by penicillia. The inset shows a magnified image of the fungi located in the micro-gap. (**c**) A schematic of bacteria (*E. coli*.) detection in an aqueous solution. The Si substrate was coated with antibodies specific to *E. coli*. (**d**) Transmittance spectra of the *E. coli*-coated (density of 0.078 μm^2^) quartz substrate in aqueous (red line) and ambient (black line) conditions. (**e**) THz transmission before (blue line) and after (red line) the deposition of E. coli on the functionalised metamaterials in aqueous environments. Inset shows the dark-field microscopic image after the deposition of *E. coli*. (**f**) THz transmission before (blue line) and after (red line) the deposition of *E. coli* on the sensor without the antibody functionalisation. Inset shows the dark-field microscopic image after the deposition of *E. coli*. [60] Copyright © 2014 Springer Nature, S. J. Park, et al.

**Figure 4 sensors-22-06896-f004:**
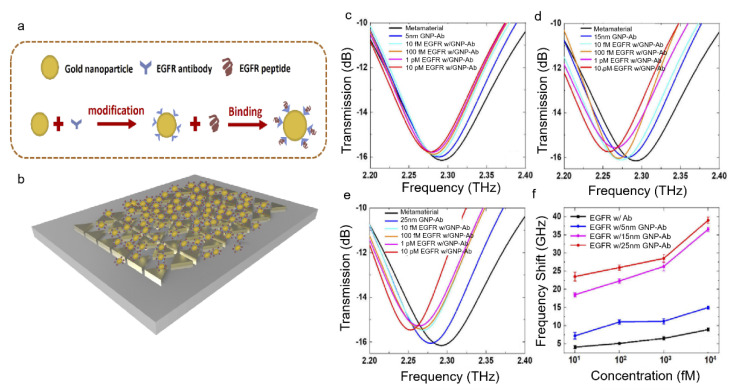
(**a**) Chemical binding of the antibody (Ab) molecules on the metamaterial surface. (**b**) The physical structure of the sensor with AuNPs introduced on the bow-tie structure. (**c**) The transmission spectra of a 5 nm AuNP-Ab functionalised sensor in response to EGFR. (**d**) The transmission spectra of a 15 nm AuNP-Ab functionalised sensor in response to EGFR. (**e**) The transmission spectra of a 25 nm AuNP-Ab functionalised sensor in response to EGFR. (**f**) Shifts in the resonance frequency for Ab functionalised bare metal structure (black line), 5 nm AuNP-Ab (blue line), 15 nm AuNP-Ab (purple line) and 25 nm AuNP-Ab (red line) functionalised metal structures [65] © 2021 Optical Society of America.

**Figure 5 sensors-22-06896-f005:**
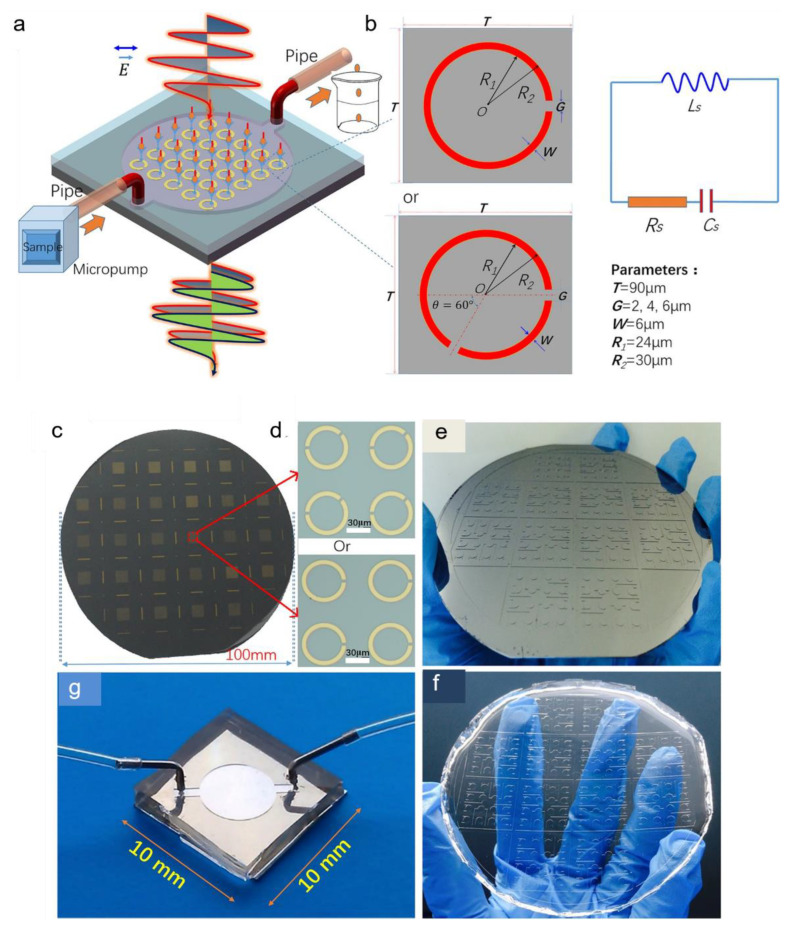
(**a**) The sketch of THz SRR biosensor integrated with a microfluidics chip. (**b**) RLC equivalent circuit of the SRR. (**c**–**g**) Fabrication steps of the microfluidics integrated SRRs. (**c**) SRRs on a 4-inch silicon wafer. (**d**) Four units of the SRR structure with one or two gaps. (**e**) The SU-8 mould for microfluidics chip fabrication. (**f**) PDMS microchannel. (**g**) The final biosensor chip integrated with microfluidics. © 2017 Springer Nature, Zhaoxin Geng, et al. [71].

**Figure 6 sensors-22-06896-f006:**
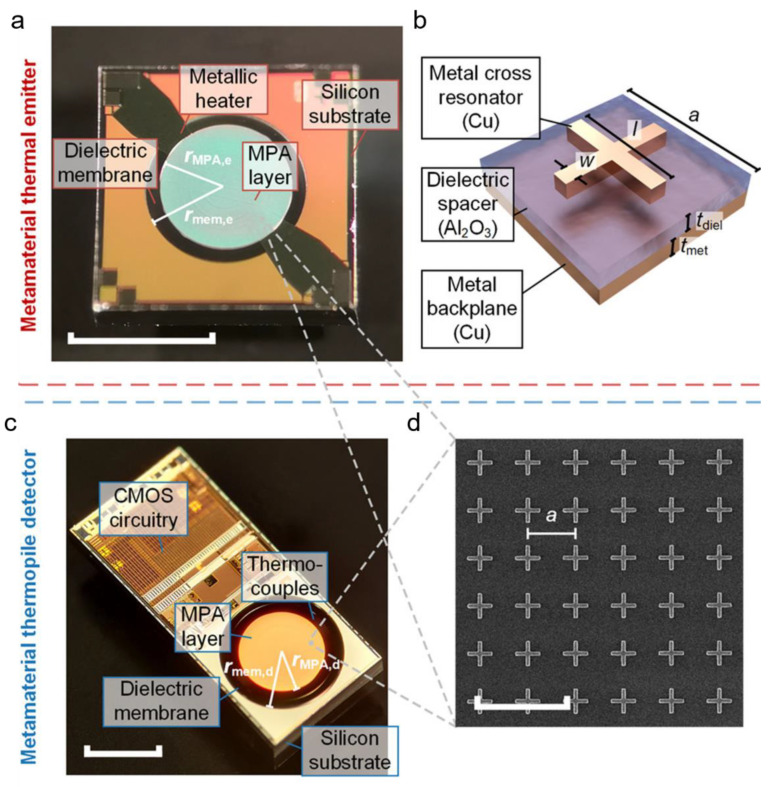
Metamaterial components. (**a**–**d**) Metamaterial thermal emitter. (**a**) The MPA layer (r_MPA,e_ = 450 μm) is fabricated on a suspended dielectric membrane (r_mem,e_ = 550 μm) with an integrated metallic heater (scale bar: 1 mm). (**b**) MPA unit cell. (**c**) The MPA layer (r_MPA,d_ = 540 μm) is post-processed on top of a suspended dielectric membrane (r_mem,d_ = 700 μm) with embedded thermocouples (scale bar: 1 mm). CMOS circuitry for signal amplification and processing is integrated on the same substrate. (**d**) Scanning electron micrograph of 6 × 6 MPA unit cells (scale bar: 4 μm) [93]. Copyright © 2020 American Chemical Society.

**Figure 7 sensors-22-06896-f007:**
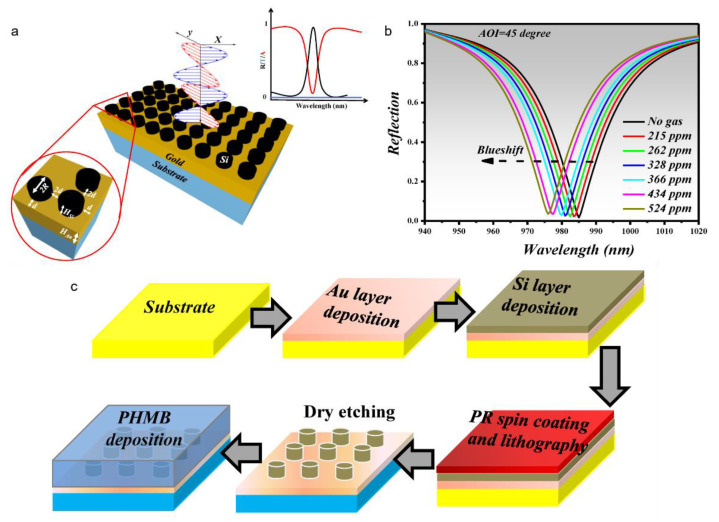
(**a**) A thin layer of PHMB functional layer was deposited on the perfect absorber metasurface. CO_2_ gas was detected via the wavelength interrogation method. (**b**) Reflection spectra for different gas concentrations in the ppm range. (**c**) Proposed fabrication scheme [97] Copyright: ©2021 MDPI.

**Figure 8 sensors-22-06896-f008:**
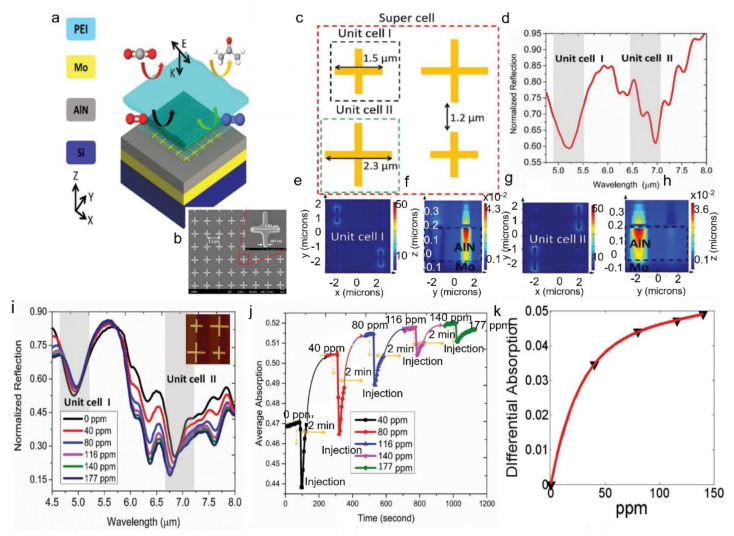
(**a**) Proposed ‘hybrid metamaterial’ absorber integrated with thin-film membrane for selective gas sensing. (**b**) FESEM (field emission scanning electron microscopy) image of the metamaterial array (inset shows the unit cell). (**c**) Design of the supercell composed of unit cell I and unit cell II for multiplexed sensing. (**d**) Simulated reflection spectrum of the supercell geometry overlaid by the gas-selective layer. Polarisation of the incoming radiation is fixed along y-axis (**e**), (**f**) electric field (at xy plane) and magnetic field (at xz plane) distribution, respectively, when unit cell I is resonant at 5.25 µm. (**g**,**h**) Electric field (at xy plane) and magnetic field (at xz plane) distribution, respectively, when unit cell II is resonant at 6.75 µm. (**i**) Steady-state sensing characteristics of the multiplexed platform. Inset: fabricated superpixel. (**j**) Dynamic absorption behaviour within the spectral window: 5–8 µm. (**k**) Differential absorption at steady state fitted under two parameter exponential models (f(x) = a ∗ exp(b ∗ x) + c ∗ exp(d ∗ x)) indicating saturation behaviour of the sensor as the gas concentration increased in continuous mode. The fitting values of a, b, c and d are 0.04555, 0.0004665, −0.04551 and −0.0691, respectively, with R-square value of 0.9993 [100]. © 2018. Published by WILEY-VCH Verlag GmbH & Co. KGaA, Weinheim.

**Figure 9 sensors-22-06896-f009:**
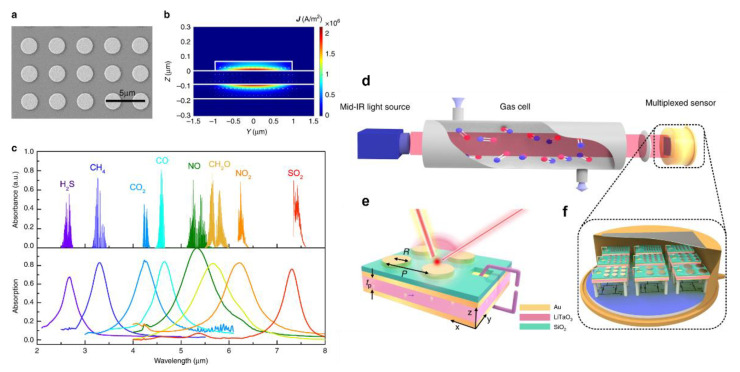
The spectral and near-field properties of the plasmonic metamaterial absorber. (**a**) Scanning electron microscope (SEM) image of the gold nanodisk antenna array. (**b**) The distribution of light-induced current density magnitude |J| and current density vector J in the yz cut-plane of the MIM absorber. (**c**) The measured absorption spectra of eight fabricated MIM absorbers and the infrared absorption bands of eight target gases: H_2_S, CH_4_, CO_2_, CO, NO, CH_2_O, NO_2_ and SO_2_. (**d**) Schematic diagram of the gas sensing system based on the proposed NDIR architecture with an array of narrowband PMA-integrated pyroelectric elements used as the spectral sensor. The method comprises three modules: a broadband light source, a gas cell and a multi-element sensor, together with necessary focusing optics. (**e**) The geometry of the narrowband detection element. From top to bottom are the Au nanodisk antenna, the silicon dioxide spacer, the gold backplate that is also used as the top electrode of the pyroelectric element, the lithium tantalate (LT) substrate and the bottom gold electrode, (**f**) the integrated package of the multiple pyroelectric elements with different detection wavelengths. [103] © Springer Nature 2020, Nature Communication.

**Figure 10 sensors-22-06896-f010:**
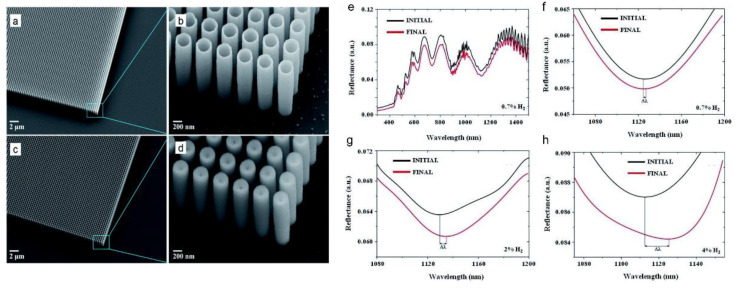
Scanning electron microscope (SEM) images of the fabricated AZO (**a**,**b**) nanotube and (**c**,**d**) pillar structures with a pitch of 400 nm, the diameter of 300 nm and height of 2 µm. The wall thickness of nanotubes is approximately 20 nm. (**e**) The response of the AZO nanotubes before and after intercalation of 0.7% H_2_ gas and upon exposure to (**f**) 0.7% H_2_, (**g**) 2% H_2_ and (**h**) 4% H_2_ gas at the wavelength range of λ = 300–1500 nm [114] © The Royal Society of Chemistry 2020.

**Figure 11 sensors-22-06896-f011:**
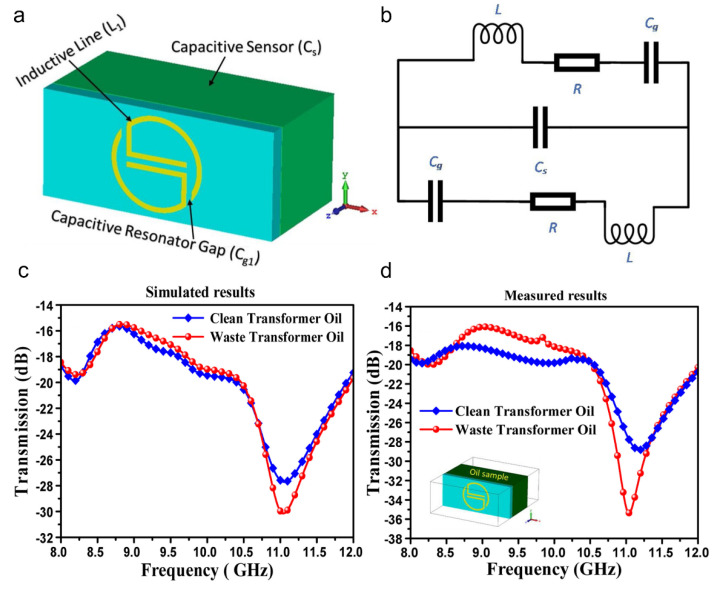
(**a**) Design of the inductive and capacitive components in the proposed structure. (**b**) Equivalent circuit diagram of the proposed metamaterial-based sensor. (**c**) Simulation and (**d**) experimental transmission spectra in response to clean and waste transformer oils [127] Copyright © 2022 Elsevier BV.

**Figure 12 sensors-22-06896-f012:**
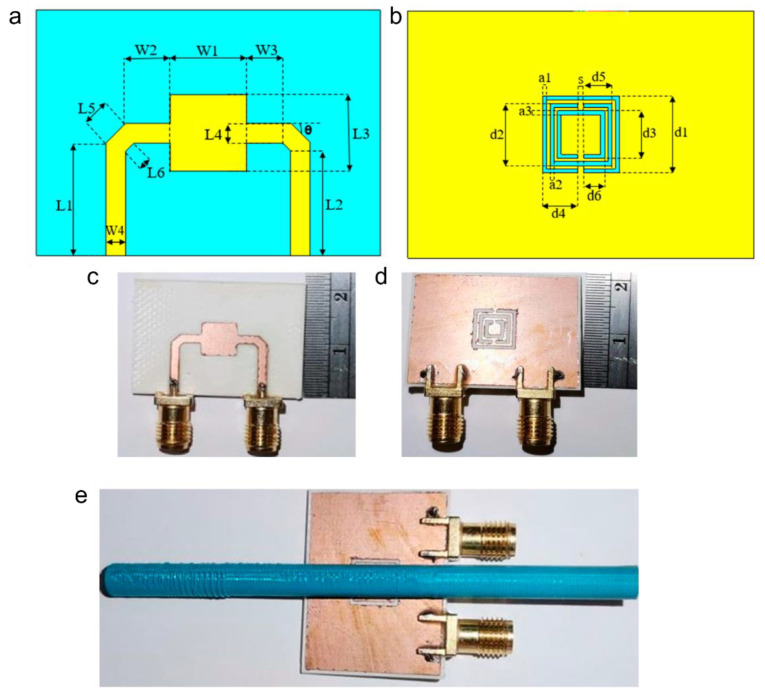
Geometric dimensions of the flat sensor structure (**a**) top view; (**b**) bottom view. S21 parameter simulations. Fabricated sensor prototype: (**c**) top view; (**d**) bottom view; (**e**) flat sensor with embedded tube [129] Copyright: © 2022 MDPI.

**Figure 13 sensors-22-06896-f013:**
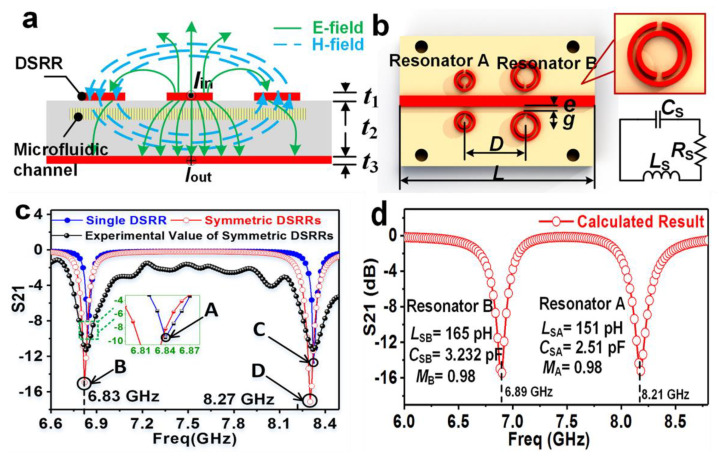
Electrical analysis of the MIM sensor. (**a**) Cross-section of a microstrip transmission line with a pair of DSRR and its electric field (E-field) and magnetic field (H-field) distribution (Thickness: t_1_ = 3.5 μm, t_2_ = 0.609 mm, t_3_ = 3.5 μm). (**b**) The equivalent circuit of DSRR. (**c**) The calculated results from the equivalent circuit of the microfluidic sensor by ADS software. (**d**) The simulated and experimental S21 spectrum of single or symmetric DSRRs. Points A and B represent resonances excited by single and dual resonator B, respectively. Points C and D represent resonances excited by single and dual resonator A, respectively [133] Copyright © 2018, Nature, Scientific Reports.

**Figure 14 sensors-22-06896-f014:**
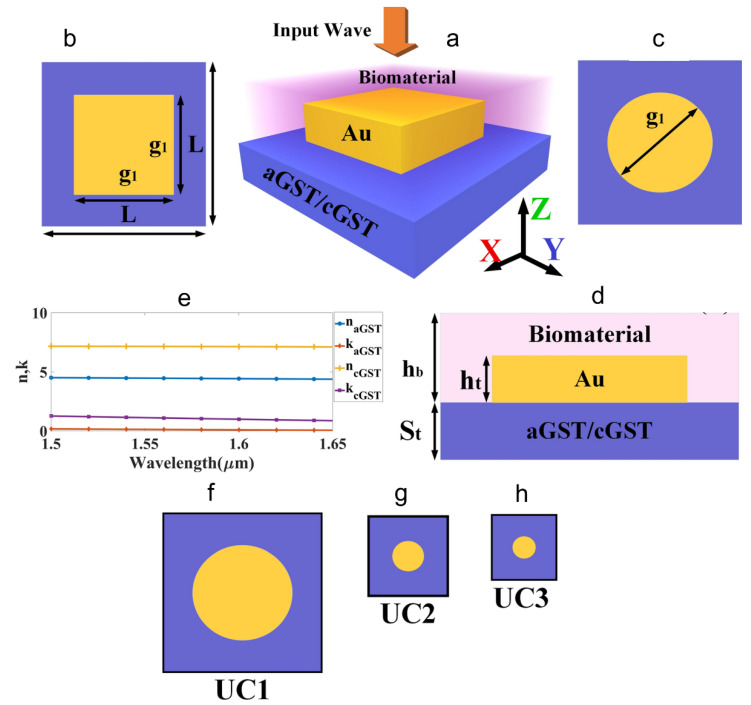
Schematic diagram of the GST-assisted metamaterial-based cubic and cylindrical resonators. (**a**) A 3D view of the sensor. (**b**) Top view of the metamaterial cubic resonator. (**c**) Full view of the metamaterial cylindrical resonator. (**d**) Side view of the sensor. The bottom layer of the structure is made of aGST/cGST. The biomolecule attaches to the top of the sensor. The incident wave is excited along the z-axis. Dimensions of the structure are S_t_ = 800 nm, h_t_ = 600 nm, h_b_ = 2000 nm, g_1_ = 1400 nm and L = 2000 nm. (**e**) Real and imaginary components of the refractive index of aGST and cGST are in the range of 1.5–1.65 µm. (**f**) The dimension of UC1 is L × L = 2000 × 2000 nm^2^, g_1_ = 1400. (**g**) The dimension of UC2 is L × L ≈ 666 × 666 nm^2^, g_1_ ≈ 466. (**h**) The dimension of UC3 is L × L ≈ 400 × 400 nm^2^, g_1_ ≈ 280 [134] Copyright © 2021, Nature, Scientific Reports.

**Figure 15 sensors-22-06896-f015:**
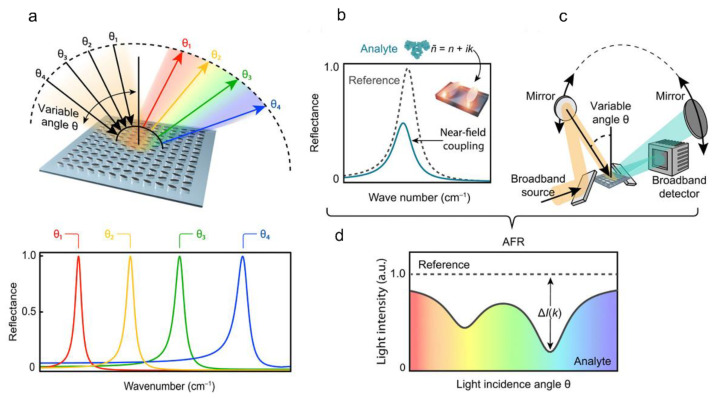
Angle-multiplexed broadband fingerprint retrieval. (**a**) A germanium-based high-Q all-dielectric metasurface delivers on-demand resonances at a specific resonance frequency n for every incidence angle q with broad spectral coverage. Continuous scanning of the incident angle produces a multitude of resonances over a target fingerprint range, realising an angle-multiplexed configuration ideally suited for surface-enhanced mid-IR molecular absorption spectroscopy. (**b**) Strong near-field coupling between the dielectric resonators and the molecular vibrations of the analyte induces an apparent attenuation of the resonance lineshape correlated with the vibrational absorption bands. (**c**) Angle multiplexing combined with the spectral selectivity of high-Q resonances allows for broadband operation and straightforward device implementation. (**d**) The chemically specific output signal of the device scheme in (**c**), which is determined by the imaginary part k of the analyte’s complex refractive index n~ (a.u., arbitrary units) [135] Copyright © 2019, Science Advance, Leitis, et al.

**Table 1 sensors-22-06896-t001:** Performance comparison of metamaterial-based biosensors.

References	Advantages/Disadvantages	Target Analyte	Sensor Configuration	Sensitivity	Frequency (f)/Analyte Concentration Range/Limit of Detection (LOD)
[21]	(+) Easy to fabricate at low cost.	biotin and streptavidin	Copper (Cu), Nickel (Ni), and gold (Au) printed on PCB	-	f range: 10.64 GHz to 10.84 GHz
[22]	(+) Low cost and easy inject printing-based fabrication	No specific analyte stated	Ag nanoparticles on paper and plastic substrate.	-	f range: 0.1 THz to 0.5 THz
[24]	(+) Minimal number of virus particles can be detected efficiently (-) Sophisticated e-beam lithography was used to fabricate the structure	60 nm of PRD1 virus and 30 nm of MS2 virus	Metamaterial structure formed by 3 nm-thick Cr followed by 97 nm-thick gold	6 GHz⋅μm^2^/particle to 80 GHz⋅μm^2^/particle	f range: 0.5 THz to 1.5 THz
[25]	(+) Faster detection in both air and aqueous environments (+) Can detect small number of microorganisms (-) Sophisticated e-beam evaporation-based metal deposition and photolithography	Yeasts and Escherichia coli BL21 (DE3), Neurospora sitophila (neurospora) and Aspergillus niger (niger)	Cr (2 nm) and Au (98 nm) metal films deposited on Si substrate	~11.6 GHz/number density	f range: 0.5 THz to 3 THz LOD: 10^7^ units/mL
[29]	(+) Higher sensitivity for four LC resonator-based SRRs as compared to a single LC resonator	Bovine Serum Albumin	Aluminium layer deposition by metal evaporation method	85 GHz/RIU	f range: 0.2 THz to 1.2 THz LOD: 1.5 μmol/L
[30]	(+) Enhanced sensitivity by adding AuNPs	The epidermal growth factor receptor (EGFR) antibody	Cr (20 nm) and Au (100 nm) bilayer film coated with AuNPs and arranged in a bow-tie configuration	1.5 to 3.9 GHz/pM	f range: 2.2 THz to 2.4 THz Conc. Range: 10 fM to 10 pM
[33]	(+) Better performance because of the SiN_x_ Film as compared to the bare Si substrate	Doped and undoped protein thin films (silk fibroin)	200 nm gold patterned on 400 nm thick SiN_x_ film deposited on Si wafer	4.05 × 10^−2^ GHz/nm.	f range: 0.1 THz to 1.2 THz
[34]	(+) High Q factor	Alpha-fetoprotein (AFP) and Glutamine transferase isozymes II (GGT-II)	The 200-nm thickness of gold on the Si wafer	3.8 GHz/(mu/mL) for GGT-II and 562.6 GHz/(μg /mL) for AFP	f range: 0.4 THz to 1.2 THz
[39]	(+) FOM > 330 (+) Sensitivity several folds higher than the conventional plasmonic sensor	Streptavidin-biotin	Au nanorod on alumina matrix	>30,000 nm/RIU	f range: 200 THz to 749 THz LOD: 300 nM
[40]	(+) high FOM of 590 (+) capable of detecting lower molecular-weight (<500 Da) biomolecules	Biotin, BSA	gold–Al_2_O_3_ and grating-coupled hyperbolic metamaterial structure	30,000 nm/RIU	f range: 150 THz to 600 THz Conc. Range: 10 pM to 1 µM

**Table 2 sensors-22-06896-t002:** Performance comparison of metamaterial-based gas sensors.

References	Advantages/Disadvantages	Target Analyte	Sensor Configuration	Sensitivity	Frequency (f)/Analyte Concentration Range/Limit of Detection (LOD)
[56]	(+) Hot spot region to enhance the plasmonic molecular coupling and improve sensitivity	CO_2_ and C_4_H_10_	A gap between two gold electrodes	2.92 × 10^−4^ ppm^−1^.	f range: 60 THz to 150 THz Conc. range: 20 to 388 ppm LOD: 20 ppm
[58]	(+) compact (+) sensitive (+) Energy-efficient gas detection (+) cascading the spectral responses of MPAs on the emitter and the detector to match the narrow absorption band of the target gas (+) highly scalable due to monolithic integration of MPAs into CMOS devices	CO_2_	Gold-coated Si spacer on a PCB board	22.4 ± 0.5 ppm·Hz^−0.5^	Conc. range: 0 to 5000 ppm
[62]	(+) Wide detection range (-) Limited to numerical analysis and lacks physical implementation	CO_2_	Nano-cylindrical meta-atoms on a gold layer deposited on a quartz substrate	17.3 pm/ppm	f range: 294 THz to 319 THz Conc. range: 0 to 524 ppm
[63]	(+) Fabricated by a low-cost CMOS MEMS technology (+) A high-quality factor of 15.7 (+) features temperature-stable and angular-independent emission characteristics (+) a 5-fold increase in relative sensitivity compared to the conventional blackbody emitter	CO_2_	a cross-shaped top Cu resonator was separated from a Cu backplane by means of a dielectric spacer layer (Al_2_O_3_)	1.7 × 10^–4^ %/ppm	Conc. range: 0 to 50,000 ppm
[65]	(+) Two wavelength-based dual-mode multiplexed gas sensing (+) fast response time (≈2 min)	CO_2_	polyethylenimine (PEI) polymer spun coated on AlN-Mo-Si	500 nm/RIU	Conc. range: 0 to 177 ppm LOD: 40 ppm
[68]	(+) Multiplexed sensing of gases in a mixture	H_2_S, CH_4_, CO_2_, CO, NO, CH_2_O, NO_2_, SO_2_	From the top to the bottom are: Au nanodisk antenna, the 80 nm silicon dioxide spacer, the Au backplate, the 75 µm lithium tantalate (LT) substrate and the 100 nm Au bottom electrode	Not stated	Conc. range: 0 to 20,000 ppm LOD: 489, 63, 2, 11, 17, 27, 54 and 104 ppm for H_2_S, CH_4_, CO_2_, CO, NO, CH_2_O, NO_2_ and SO_2_
[72]	(+) Highly reliable, re-usable and selective (+) a new signature evolving at 300 MHz	NO_2_	Fe_3_O_4_ nanoparticles on two square ring-shaped slots	0.2 MHz/ppm	f range: 200 MHz to 800 MHz Conc. range: 0 to 110 ppm
[75]	(+) the presence or absence of H_2_ can be monitored by direct visual inspection (+) response time of only 10 s (+) low-cost fabrication using a simple electrochemical technique	H_2_ and N_2_	Bimetallic Au/Pd nanorod on a glass substrate	-	f range: 333 THz to 750 THz LOD: 1% H_2_
[79]	(+) large sensing area (+) high sensitivity at room temperature (+) fast response in 10 min (-) sophisticated ion reactive etching and atomic deposition layer	H_2_	Aluminium-doped Zinc oxide (AZO nanotubes) on SiO_2_/Si substrate	0.0006 a.u./%	f range: 250 THz to 333 THz Conc. range: 0.7 to 4% LOD: 0.7%

**Table 3 sensors-22-06896-t003:** Performance comparison of metamaterial-based chemical sensors.

References	Advantages/Disadvantages	Target Analyte	Sensor Configuration	Sensitivity	Frequency (f)/Analyte Concentration Range/Limit of Detection (LOD)
[91]	(+) real-time (+) fast (+) low cost (+) durable (+) accurate detection	Clean and waste transformer oil, Corn, olive and cotton oils, branded and unbranded diesels, aniline-doped ethyl-alcohol and benzene-doped carbon tetrachloride	Copper pad on both front and backside of FR-4 substrate	250 MHz/ 0.11 ε_r_	f range: 8 GHz to 12 GHz LOD: Not stated (detection was based on separation of resonance peaks)
[93]	(+) Linear relationship between pesticide concentrations and transmission amplitudes	2,4-dichlorophenoxy acetic and chlorpyrifos solutions	multiwalled CNT arrays on a silicon substrate	1.38 × 10^−2^/ppm from 1–10 ppm and 3.0 × 10^−3^/ppm over 10 ppm	Conc. range: 1–10 ppm and 10–80 ppm
[97]	(+) improved sensitivity due to the integration of inter-digital capacitor (IDC) topology (+) better frequency resolution compared to existing SRRs (+) simple design (+) easy fabrication (+) economical	Glucose	Copper SRR made on Rogers RT6006 substrate and integrated with PDMS microfluidic channel	0.026 MHz/(mg/dL)	f range: 3 GHz to 5 GHz Conc. range: 0–5000 mg/dL
[98]	(+) miniaturised (24*15*0.6 mm^3^) (+) reusable (+) label-free (+) non-destructive (+) smaller sample volume (4 µL) (+) multi-band sensing (+) better linearity in ethanol sensing (−2.80%)	Ethanol-water mixture	Copper coated with 3.5 µm thick Ni/Au layer on Rogers 4003c substrate (0.203 mm thick)	2.1 × 10^6^ Hz/%	Conc. range: 0–100% of ethanol in water-ethanol mixture
[99]	(+) Tunable response	Haemoglobin, urine	amorphous GST (aGST) and crystalline GST (cGST) in different design structures	825–1795 nm/RIU when tested on haemoglobin, and 1000–2333 nm/RIU when tested on urine	f range: 181 THz to 200 THz Conc. range: 10–40 g/L for haemoglobin, and 0–10 mg/dL for urine
[101]	(+) Optimised asymmetric electric split-ring resonator (AESRR) topology (+) distinguish liquids and solid dielectric materials with bigger frequency shift and higher sensitivity. (+) low-cost (+) real-time (+) high sensitivity (+) high robustness	Peanut oil, Corn oil, Sunflower seed oil, Soybean oil, Isopropyl alcohol, ethyl acetate, ethanol	Copper pad on FR-4 substrate	0.612	LOD: Not stated (detection was based on separation of resonance peaks)
[102]	(+) compact design on a single PCB (+) low cost (+) contactless (+) reusable (+) easy to fabricate	Ethanol–water mixture	Copper pad on FR-4 substrate	0.57	Conc. range: 0–100% of ethanol in water-ethanol mixture
[103]	(+) high sensitivity detection of scattered data (+) adequate penetration depth	Glucose	Copper pad on FR-4 substrate	0.0125 dB/(mg/dL)	f range: 2.2 GHz to 3.8 GHz Conc. range: 100–300 mg/dl
[104]	(+) Ultralow limit of detection	anti-BSA	Al coated periodic nanopillar arrays	0.14 ng/mL	f range: 333 THz to 1000 THz Conc. range: 0.001–1000 ng/mL LOD: 1 pg/mL
[105]	(+) No pretreatment required	Vitamin D	Au coated cross and star shaped nanostructures on silicon substrate	500–800 nm/RIU	LOD: 86 pM

## Data Availability

Not applicable.

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
