# Peer review of "Metasurfaces for Sensing Applications: Gas, Bio and Chemical"

_sensors, 2022, doi:10.3390/s22186896_

Round 1

Reviewer 1 Report

The manuscript is a comprehensive review of metasurfaces based biosensors presenting the fundamental aspects and bio-sensing, chemical and gas sensing applications. The field of metamaterials for sensing is an emerging field, and this review will be helpful for the interested readers to get a collective idea in this area. However, the field is so wide with continuous developments everyday that it is impossible to include all the works in a single review and all the reviews in the field (and this one as well) cover a small niche only. Authors did a good job to achieve a sound and well readable review of many activities in the field. 

I have a few minor comments and recommendations.

1.     I could not get the logic behind the figure and table numbers (Fig 4.1….. means what?). Either it should be according to the section number or it can be kept common for the full article.  

2.     There is no mention of table in the manuscript. The description of tables in the text should be included for completeness.

3.     The author should include more examples of SPR and LSPR sensors for comparison in introduction section, for eg. Biosens. Bioelectron. 196, 113720 (2022).

4.     In general, transitions from section to section should be made smoother and better justified sticking to the narrower scope of the review.

5.     In section 3.1 and 3.2 the subheadings would be better justified if authors could provide some introduction at the beginning for eg. “The sensors are subdivided in the sections according to the type of metasurfaces used” for clarity purposes. Specifically in section 3.2 Gas sensing, only one type of gas is discussed in such section, in that case the subheadings should be justified.

6.     Small comment on figures is related to poor resolution in figure 4.1.4 and 4.2.3

  1. Currently, a critical comparison of advantages and drawbacks as well as future trends of such technology is missing in the paper. The review should not only describe the literature works but also provide a critical analysis at the end. The authors should elaborate more on such aspect.

Author Response

I have attached the file.

Reviewer 2 Report

The review manuscript entitled “Metasurfaces for sensing applications: gas, bio and chemical” by authors Tabassum et al. focuses the advances in the field of metamaterials and metasurfaces. The manuscript perfectly matches the criteria of the journal and need of beginner researchers starting their research in the field of metamaterials and plasmonics. The authors have covered the applications of meta surfaces in various fields especially gas, bio and chemical sensing in detail. The review article is written very well and in an organized way. There are a few minor comments which can be included before publication in the journal “Sensors”.

1.    Authors must avoid the use of abbreviations in abstract.

2.    Some abbreviations are used without their full form for the first time e.g. MA, LC. RCA etc.

3.    In the entire manuscript, figures numbering is not clear.

4.    Few figures have very low resolution for example Fig. 4.1.4. These figures need to be in adequate quality.

Author Response

I have attached the file.

Reviewer 3 Report

My comments are mentioned are mentioned in the attached PDF file. One additional critical question/suggestion is to comment/add some sentences on the possible capability of the mentioned sensors for continuous monitoring of gases/chemical entities.

Author Response

I have attached the file.
